# Valuation of Ecosystem Services for the Sustainable Development of Hani Terraces: A Rice–Fish–Duck Integrated Farming Model

**DOI:** 10.3390/ijerph19148549

**Published:** 2022-07-13

**Authors:** Yuan Yuan, Gangchun Xu, Nannan Shen, Zhijuan Nie, Hongxia Li, Lin Zhang, Yunchong Gong, Yanhui He, Xiaofei Ma, Hongyan Zhang, Jian Zhu, Jinrong Duan, Pao Xu

**Affiliations:** Key Laboratory of Freshwater Fisheries and Germplasm Resources Utilization, Key Laboratory of Integrated Rice-Fish Farming Ecology, Ministry of Agriculture and Rural Affairs, Freshwater Fisheries Research Center, Chinese Academy of Fishery Sciences, Wuxi 214081, China; yuany@ffrc.cn (Y.Y.); xugc@ffrc.cn (G.X.); shennn@ffrc.cn (N.S.); neizj@ffrc.cn (Z.N.); lihx@ffrc.cn (H.L.); zhangl@ffrc.cn (L.Z.); gongyc@ffrc.cn (Y.G.); heyh@ffrc.cn (Y.H.); maxf@ffrc.cn (X.M.); zhanghy@ffrc.cn (H.Z.); zhuj@ffrc.cn (J.Z.)

**Keywords:** valuation, ecosystem services, Hani terraces, rice–fish–duck integrated farming, sustainable development

## Abstract

As a complementary and symbiotic agro-ecological cycle system, a nature-based integrated rice–fish–duck farming ecosystem was developed in the Honghe Hani Rice Terraces. The main research objective was to evaluate the ecosystem services based on case studies of the Hani integrated rice–fish–duck terraced farming system and determine its potential and its importance as an ecological asset. We developed a valuation model to assess the value of the integrated farming system based on the three aspects of provisioning, regulation and maintenance, and cultural services; we selected eight groups and 10 indictors to evaluate the ecosystem services of the integrated ecosystem in Honghe Hani Rice Terraces was 3.316 billion CNY, of which the provisioning service value was 1.76 billion CNY, the regulation and maintenance service value was 1.32 billion CNY, and the cultural services value was 230.85 million CNY. The evaluation will be useful as a theoretical reference for poverty alleviation policy makers in similar poverty-stricken areas, enabling them to better protect and promote this mode of farming and further promote the protection of the natural environment and cultural heritage alongside the sustainable development of natural resources and human well-being.

## 1. Introduction

The concept of ecosystem services (ES) first attracted the attention of academia in the 1970’s (MIT, 1970; Westman, 1977) [1,2]. In the 1990s, there was a general realization that a systematic approach was required to evaluate the relationship between humans and nature (Schulze, 1993; Daily, 1997) [3,4]. To effectively manage the frequent occurrence of disasters caused by climate change, water shortages, and environmental crises, as well as threats to human survival and sustainable socio-economic development, the United Nations launched the Millennium Ecological Assessment Project in 2000 to promote the awareness and popularity of ecosystem services (Powledge, 2016) [5], taking into account the increasing human demand for ecosystem services (MEA, 2005) [6]. Subsequently, the concept of nature-based solutions was developed, with innovative solutions proposed to protect biodiversity and promote sustainable development, while mitigating and adapting to the impacts of climate change (World Bank, 2008) [7].

Integrated agro-aquaculture farming has become widely used as a sustainable and nature-based solution to food production. Fish farming in rice fields is a traditional Chinese ecological agricultural model, with a history dating back more than 1300 years (Lu and Li, 2006; Liu et al., 2015) [8,9]. This model places rice and fish in the same field, which not only utilizes the mutualistic symbiosis between rice and fish to achieve the purpose of ecological farming, but can also continuously produce food crops and freshwater products (Stankus and Halwart, 2017) [10] and enhances biodiversity conservation (Frei and Becker, 2005) [11]. The service of the integrated agro-aquaculture system can be divided into three categories. The first is the provisioning service, which refers to the conditions of the natural environment and the utility it provides for human survival through the combination of ecological processes in rice fields and human activities. The second is the ecological service, which includes actions such as maintaining and improving the atmospheric regulation of the natural environment, soil and water conservation, and purifying the environment (Agus et al., 2006) [12]. The third is the social service, which includes actions such as promoting rural economic development and rural area vitality, cultural protection, social security, and food security (Costanza R, 2017) [13].

The Honghe Hani Rice Terraces in Yunnan are famous for their widely distributed rice terrace landscape, especially the well-preserved traditional agricultural system. The terraces have also played an important wetland function in regulating climate, preserving water and soil, and maintaining biodiversity (Tsuruta et al., 2011) [14]. In 2010, the Yunnan Honghe Hani rice terrace system was designated by the United Nations Food and Agriculture Organization as a Globally Important Agricultural Cultural Heritage (GIAHS) (Yuan et al., 2014) [15]. In 2013, it was selected in the first batch of China’s important agricultural cultural heritage (China-NIAHS) sites by the Ministry of Agriculture, and it was successfully included in the United Nations Educational, Scientific and Cultural Organization (UNESCO) World Cultural Heritage List in the same year (Chan et al., 2016) [16]. However, the economic, scientific and aesthetic values of the Hani terraces were not fully realized at the time due to a lack of systematic, effective protection and management. The profits generated by farming the terraces were low compared with other income-generating activities, and most of the local young and mid-aged labor force chose to work elsewhere while a few others took up dry farming, which was relatively more profitable. As a result, many paddies were abandoned and became wasteland, bringing devastating changes to the original appearance, cultivation traditions, and ecosystems of the Hani terraced fields. To maintain the unique ecosystem of the Hani terraces, which cover a complex range of ecosystems incorporating forests, rice fields, water bodies, and villages, since 2015 the Hani local government has encouraged local farmers to carry out integrated rice–fish–duck faming over the whole Hani terraced area.

Academic studies of ecosystem services have mainly focused on four aspects. (1) Research on the concept (Costanza et al., 1997, 2014, 2017) [13,17,18], connotation (Daily, 1997) [4], and classification system of ecosystem services (Fisher and Turner, 2008; Reid and Mooney, 2016) [19,20], which has attempted to quantify and systematize the scientific theory of ecosystem service value. (2) Research on the changing mechanisms of ecosystem services and their interaction with biodiversity (Wang et al., 2013; Pascual et al., 2017; Wood et al., 2018) [21,22,23]. (3) Research on the evaluation techniques used to value ecosystem services (He et al., 2018; Hardaker et al., 2020) [24,25]. (4) The use of different methods to evaluate a single ecosystem or single service in different regions (Richards and Tunçer, 2018; Schirpke et al., 2018; Taffarello et al., 2020; Castonguay et al., 2014, 2016; Burkhard et al., 2015; Horgan et al., 2016, 2017; Dang et al., 2018a, 2018b, 2019; Gu et al., 2012; Liechti and Rodewald., 2020 and Jiao et al., 2019; Jiao et al., 2014 focus on Hani terrace) [26,27,28,29,30,31,32,33,34,35,36,37,38,39,40]. Ecosystem service value evaluations have evolved from a single quantitative calculation to more dynamic, mechanistic, and application-based comprehensive research. These previous studies provide a theoretical basis for the present analysis. Most previous studies have focused on the protection and development of cultural heritage, and only a few studies have attempted to evaluate the ecosystem service value of the integrated agro-aquaculture ecosystems in the Honghe Hani Rice Terraces.

As a complementary and symbiotic agro-ecological circular system, it aims to ensure improvements in the output of rice, and the quality of rice, fishery and poultry products to increase output and add value to terraced fields, and to unify economic, social, and ecological benefits. Although the integrated rice–fish–duck farming practiced in the Hani terraces has these functional values, there is no perfect system to express these values in monetary terms, which affects the choice of land use methods. Against this background, the establishment of an evaluation system for the Hani terraces could enable its value to be realized and form the basis for terraced field resource planning. An evaluation of the ecosystem services provided by the integrated rice–fish–duck ecosystem will also provide a research basis for the establishment of an ecological compensation mechanism, which can protect the inheritance and development of the world agricultural cultural heritage, realize and improve human well-being, and create a better natural environment to meet ecological needs, providing new ideas for the sustainable development of people and nature.

This study considered integrated rice–fish–duck farming in the Honghe Hani Rice Terraces as a case study. It adopted the market price, replacement costs, and equivalence factor methods to establish the value of provisioning services, gas regulation, climate regulation, air purification, pest control, maintaining biodiversity, water regulation, soil conservation, soil organic accumulation, and cultural services, i.e., a total of 10 service value analysis indicators. A driving force, pressure, state, impact, and response (DPSIR) analysis model was developed to analyze the driving factors and responses of the Hani terraces rice–fish–duck ecosystem more comprehensively and systematically. The potential future social and economic developments were also considered, as well as their contribution to the achievement of sustainable development and nature-based solutions for terraced fields in other plateau areas of the world.

## 2. Materials and Methods

### 2.1. Study Area

#### 2.1.1. Geographic Location

Honghe Hani and Yi Autonomous Prefecture is located in the southeast of Yunnan Province and is traversed in the east-to-west direction by the Tropic of Cancer. The prefecture has a total area of 32,931 km^2^, with four county-level cities, six counties, and three autonomous counties under its jurisdiction (Shu et al., 2021) [41]. The Hani terraces have a history dating back more than 1300 years (Bai, 2013) [42] and are located in the south of Honghe Hani and Yi Autonomous Prefecture (22°26′–23°26′ N, 101°48′–103°38′ E). They include the four counties of Honghe, Yuanyang, Lvchun, and Jinping (Figure 1), with a total area of about 56,666 hm^2^. This area is located to the south of the Tropic of Cancer, with an altitude of 88 to 3053 m. It has a subtropical plateau humid monsoon climate, with distinct dry and wet seasons, during which foggy and rainy weather can be experienced. The region has an annual precipitation of 1344 mm and annual sunshine of more than 2000 h (Liu et al., 2010) [43].

#### 2.1.2. Unique Topography

The Hani people have created a spectacular terraced civilization by exploiting the special geographical terrain through the concept of “as high as the mountains are, as high as the waters”. This has resulted in the development of a “forest-village-terrace field-water system” agricultural ecosystem (Figure 2), which is a composite agricultural ecosystem with an intelligent structure, complete functionality, diverse values, and strong self-regulating ability. Forests grow on the mountain tops and villages are built under the forests. Water flows from the mountain tops and is slowed by the forests, a process that is conducive to irrigation and water conservation, with the result that mountain springs and streams have water all year round. A water supply for humans, animals, and terrace irrigation is guaranteed, and the local biodiversity is preserved (Liu et al., 2014) [44]. The vertical character of the Hani terraces is an important reason for the formation of a benign ecological circular system. The construction of the terraces completely follows the contour line, which reduces the amount of earthwork used and prevents soil erosion. This structure realizes a high degree of integration between man and nature and reflects the characteristics of ecological agriculture with an intelligent structure, complete functionality, diverse values, and strong self-regulating ability (Min and Tian, 2015) [45].

#### 2.1.3. Local Method of Rice–Fish–Duck Integrated Faming

The light, climate, water source, water quality, soil, water temperature, and humidity (hardness) in the Hani terraces provide good conditions for the implementation of integrated rice–fish–duck farming. According to a survey of the Bureau of Statistics, there are 17,640 hm^2^ of terraced fields in Honghe Prefecture, including 13,687 hm^2^ of red rice planting area (with an output of 73,600 tons), and 13,373 hm^2^ of rice–fish–duck integrated farming projects. The typical circle-shaped sump in the rice field is 0.6–1.0 m deep, and 30 cm above the soil level of the paddy field (Figure 3).

### 2.2. Data Collection

Three sources of data were used in this study. First, through face-to-face questionnaire-based interviews conducted from September to December 2021, we obtained the yields and prices of rice, fish, and duck products from 100 farmers using a random sampling method. In the same period, we conducted a field survey to obtain the organic matter, total nitrogen, total phosphorus, and total potassium content in the soil tillage layer. Second, some monitoring data, such as the number of the growth periods of rice, the number of days of standing water for rice, the number of hot days in summer, the price of agricultural water and coal, the number of tourists, and the tourism revenue were obtained directly from the relevant authorities and official websites. Third, the ecological parameters of the rice–fish–duck ecosystem (water evaporation, soil permeability, and average sulfur dioxide (SO_2_), oxides of nitrogen (NOx), hydrogen fluoride (HF), and dust concentrations absorbed by the terraces were obtained from the published literature. The questionnaire (Appendix A) requested basic information regarding the farming households, including socioeconomic characteristics, asset ownership, education levels, the area of the integrated rice–fish–duck faming system, the input cost of production materials and sales, and the understanding of terrace services. Where the required data could not be obtained through field studies, the information was extracted from the published literature. The relevant data for the single-rice system was obtained from historical data provided by local government departments. All required data were shown in Table 1.

### 2.3. Construction of the Valuation Model

The Hani terrace integrated rice–fish–duck farming ecosystem produces fish and duck products alongside rice and has the functions of maintaining water and soil quality, regulating the climate, maintaining the ecological value of ecosystem biodiversity, maintaining social harmony and stability, and intelligently integrating nature (Zhan and Jin, 2015) [46]. There is also scientific value to the ecosystem, and the aesthetic value of the landscape is considerable (Jiao et al., 2012) [47]. We developed a valuation model to assess the integrated rice–fish–duck faming system based on the Common International Classification of Ecosystem Services (CICES) V5.1 (2018), which was designed to help measure, account for, and assess ecosystem services and is accepted internationally. We selected eight groups and 14 indictors from the provisioning, regulation and maintenance, and cultural service aspects, to evaluate the ecosystem services of the integrated rice–fish–duck system in the Honghe Hani Rice Terraces depending on data availability and method feasibility (Table 2).

### 2.4. The DPSIR Model of the Integrated Rice–Fish–Duck Farming Ecosystem

Model localization is an important link in the quantification-balance-decision approach when applying ES models. A DPSIR model (Weber, 2007) [48] was developed to effectively integrate the social, economic, and environmental systems of the Hani terraces (Hou et al., 2014) [49]. A DPSIR model can comprehensively and systematically analyze the interacting processes in human–environmental systems (Feld et al., 2010) [50]. Because DPSIR models can explain the cause–effect relationships between the different sectors of economic, ecological, and social systems (Kristensen, 2004) [51], they have been widely used to improve the application of ecosystem services and provide more comprehensive and systematic results. In this study, we used a DPSIR model (Svarstad et al., 2008) [52] to identify the driving factors and responses of the Hani terraces rice–fish–duck ecosystem.

The meanings of the terms used in the DPSIR model are as follows. Driving force represents the existing reasons for the development of the Hani terraces ecosystems. Pressure represents the direct impact of any existing problems on the development of the Hani terraces. State represents all the factors affecting the development of the Hani terraces. Impact represents the impact of the state on the natural environment and social economy during the integration of rice–fish–duck farming. Response represents the countermeasures taken by different stakeholders to mitigate the drivers of change.

### 2.5. Estimation of the Ecosystem Service Values of the Integrated Rice–Fish–Duck Farming Ecosystem

In the following ecosystem service value estimates, the market value and replacement cost methods for determining ecological value were used to measure the values of gas regulation, water regulation, and other aspects. Gas regulation is comprised of carbon fixation, oxygen release, and greenhouse gas emissions, and its service value was estimated by the afforestation cost method and global warming potential (GWP) (Xiao et al., 2019) [53]. The equivalence factor method was used to estimate the value of maintaining biodiversity (Xie et al., 2015) [54]. The reference parameters were converted to 2020 prices according to the price index published by the National Bureau of Statistics (https://www.ceicdata.com/zh-hans/china/agricultural-production-price-index, accessed on 1 December 2021) because price fluctuations occurred during the long research period. Table 3 shows the 10 service values and the corresponding formulas used for their calculation, with reference to previous studies (Zheng et al., 2018; Jiang et al., 2016) [55,56], together with an explanation of the parameters.

## 3. Results

### 3.1. Provisioning Service Value of the Hani Terrace Ecosystem

The provisioning service value of providing primary products, such as rice, fish, duck, and duck eggs, was estimated by the market value method. The field survey determined that the rice varieties were mainly high-yield, high-quality, disease-resistant, cold-resistant, and adaptable, such as Hongyang No. 2 and No. 3. The fish species were mainly loach and common carp. The ducklings were local species, which can generate a considerable profit. The average yield of red rice was 5370 kg/hm^2^ and the price was 6 CNY/kg, giving a total economic value of rice of 430.89 million CNY. The average output of fish (common carp) products was 750 kg/hm^2^ and the price was 50 CNY/kg, these fish being preferred by consumers due to their ecologically farmed credentials, giving a total value provided by fish products of 501.5 million CNY. Typically, there were 150 drakes and 225 female ducks/hm^2^, laying 21,600 eggs/hm^2^. The price of each duck egg was 2 CNY and the price of each duck was 50 CNY. The total value of duck products was therefore 828.48 million CNY. The rice, fish and duck are all produced from the GIAHS areas, and consumers consider this a trustworthy source. The average provisioning service value was 131,670 CNY/hm^2^, and the total provisioning service value was 1.76 billion CNY.

### 3.2. Regulation and Maintenance Service Value of the Hani Terrace Ecosystem

#### 3.2.1. Gas Regulation

The total value of gas regulation in the evaluation area was the sum of the total value of CO_2_ fixation, O_2_ release, and the value of greenhouse gas emissions. The rice yield in the evaluation area was 73,600 t. We adjusted the afforestation cost to 327.32 CNY/t C according to the forestry price index released by the National Bureau of Statistics in 2020 based on a previous study (Dahowski et al., 2009) [63]. Following the forestry industry standard of the People’s Republic of China, we adjusted the cost of industrial oxygen to 1072.21 CNY/t O_2,_ according to the price index in 2020 based on a previous study (Liu et al., 2021) [64]. The CO_2_ fixation service value was 5.35 million CNY. The O_2_ release service value was 46.95 million CNY. The total value of CO_2_ fixation and O_2_ release was 52.30 million CNY.

The greenhouse gases emitted in the evaluation area mainly comprise CH_4_ and CO_2_. The methane (CH_4_) and CO_2_ emissions in the rice field ecosystem were converted into total CO_2_ emissions.

The evaluation area was 13373.33 hm^2^, and the rice growing period in the evaluation area was generally 180 days. The average emission fluxes of CH_4_, rice CO_2_, and soil CO_2_ from rice fields in Yunnan Province were 0.334, 272.04, and 102.35 kg/(hm^2^/d), respectively. The greenhouse effect produced by 1 kg of CH_4_ was equivalent to the greenhouse effect produced by 24.5 kg of CO_2_. Total CH_4_ emissions were 585,315 kg, and total CO_2_ emissions were 788,577,460 kg, so the CH_4_ and CO_2_ emissions in the rice field ecosystem were equivalent to a total of 802,917.69 t of CO_2_ emissions. The total value of greenhouse gas emissions was 71.67 million CNY. The value of greenhouse gas emissions was greater than the combined value of CO_2_ fixation and O_2_ release, so the total value of gas regulation was −19.36 million CNY.

#### 3.2.2. Climate Regulation

The climate regulation of the Hani terraces field ecosystem was mainly reflected in the degree of cooling effect on the surrounding areas. Honghe has an average of 60 hot summer days. Its local climate is arid with abundant sunshine, and the average daily water evaporation in terraced fields is 3.83 mm/day. The total cooling effect was therefore 229.8 mm.

The quantity of heat consumed when water evaporated from terraced fields was equivalent to 140.50 tons of standard coal per year. The value of climate control, calculated using the standard coal price based on the coal price in 2020 (600 RMB/t), was 1127 billion CNY/hm^2^/year.

#### 3.2.3. Air Purification

The average SO_2_, NOx, HF, and dust concentrations absorbed by terraced fields were 45, 33.3, 0.57, and 33,200 kg/hm^2^/year, respectively (Tang, 2019) [65]. According to the forestry industry standards of the People’s Republic of China, the cost of SO_2_ removal is 1.44 RMB/kg, the cost of NOx removal is 0.76 RMB/kg, the cost of HF removal is 0.83 RMB/kg, and the cost of dust removal is 0.18 RMB/kg. Therefore, the air purification value of the Hani terraces was 6066.36 yuan/hm^2^ (2004). Based on the price index of the National Bureau of Statistics, this value was adjusted to 8968.61 CNY/hm^2^ (2020). The total air purification value of the Hani terraces was therefore 119.94 million CNY.

#### 3.2.4. Pest Control

According to our survey data and the results of Li F et al. (2021) [66], the integrated rice–fish–duck system in the Hani terraces could reduce pesticide costs by approximately 50%. The pesticide cost under the rice monoculture system was approximately 750 RMB/hm^2^/year in 2020. We used the replacement price method to calculate the pest control value of the integrated rice–fish–duck in the Hani terraces, and arrived at a figure of 375.00 RMB/hm^2^/year (Wang et al., 2020) [67]. The total pest control value was therefore 5015 million CNY.

#### 3.2.5. Maintaining Biodiversity

The equivalence factor for maintaining biodiversity in paddy fields was 0.21. Based on the weighted average income from wheat, maize, and rice, the value of biodiversity in each unit was 628 CNY/hm^2^. However, we adjusted this price to 911.98 RMB/hm^2^ (Liu et al., 2021) [64] for 2020 based on the price index. The value of maintaining biodiversity was therefore 12.20 million CNY.

#### 3.2.6. Water Regulation

The market price of agricultural water was 0.2 CNY/m^3^, and the number of days of standing water during the growth period of rice was 160 d. The water regulation value was therefore 3.09 million CNY.

#### 3.2.7. Soil Conservation

According to our investigation of the Hani terraces, the organic matter, total nitrogen, total phosphorus, and total potassium contents in the soil tillage layer were 44.61, 1.32, 0.43, and 14.9 g/kg, respectively. The price of fertilizer was 2.75 CNY/kg, according to our field survey. The soil thickness in the tillage layer was 0.2 m and the soil bulk density was 1.09 g/cm^3^. The soil conservation value was therefore 491,140 CNY.

#### 3.2.8. Soil Organic Matter Accumulation

Because of the lack of available rice and straw biomass and carbon content data for the Hani terraces, data was extracted from a study in Jingzhou City by Jiang (2016) [56]. Straw biomass was estimated to be 1.24 × 104 kg/hm^2^, the straw carbon content was estimated to be 41.4%, the rice root biomass was estimated to be 0.21 × 104 kg/hm^2^, and the root carbon content was estimated to be 34.6%. Therefore, the input of organic carbon due to straw and rice roots was 4198 kg/hm^2^.

For the output of soil organic matter (OSOC), we also extracted data from Jiang (2016) [56]. The amount of CO_2_ released from the terraced fields was 2123.63 kg/hm^2^, and the annual CH_4_ emissions were 29.64 kg/hm^2^. The amount of organic matter released from the terraced fields was 595.61 kg/hm^2^ and the net soil organic matter accumulation was 3602.09 kg/hm^2^. The market price of organic fertilizer calculated by the amount of pure carbon was 1.53 RMB/kg C based on the 2020 price index. Finally, the value of soil organic matter accumulation was determined to be 75.63 million CNY/year.

### 3.3. Cultural Services

Because it is difficult to take into account the daily travel expenses incurred when visiting the integrated rice–fish–duck farming ecosystem, the evaluation of the cultural value referred to the income generated from tourists participating in the rice-fish cultural festival in the Yuanyang and Samaba terraces. From the tourism data in the Honghe Hani and Yi Autonomous Prefecture Yearbook, it was calculated that the unit area value of Hani terrace landscape tourism in Yunnan was 17,262 CNY/hm^2^, and the value of cultural services was 230.85 million CNY.

### 3.4. Analysis of DPSIR Model of the Hani Rice–Fish–Duck Integrated Farming

Driving force of our study is the low production levels and benefits of the terraces. Because only one crop of red rice is planted every year, an annual yield of 5100 kg/hm^2^ generates a net profit of less than 6000 CNY/hm^2^ for a family, and therefore poverty remains a serious issue. Due to the specific geography and impossibility of fully mechanizing farming in the region, it is very difficult for people to farm on terraces, and considerable manpower is required for all farming tasks. Pressure facing Hani terrace development is that infrastructure construction lags behind (Figure 4), which led to the poor condition of producing and living. As a result, most of the young and strong labor force choose to work elsewhere, while those left behind to continue agricultural production in the village are mainly elders and less educated women. The Heritage protection has been affected. State represents all the factors affecting the development of the Hani terraces. Because of the driving force and pressure components, the terraces cannot be effectively managed and may even be abandoned. Some farmers have decided to switch directly to other high value dry-farmed plants, such as sugar cane, vegetables, and herbs that do not require constant watering. Therefore, conducting integration of rice–fish–duck farming was one of the impacts. Because this farming mode can potentially result in high economic returns, with good ecological and social benefits, it can ensure heritage protection, the sustainable development of the Hani terraces, and food security together with human well-being. Response represents the countermeasures taken by different stakeholders to mitigate the drivers of change, including local government development of sustainable integrated rice–fish–duck faming and provision of policy support, creating enterprises to develop a rice–fish–duck value chain, and helping research institutes to study the species involved and build collaborative innovation platforms in response to the various factors involved.

### 3.5. Comparison of ES Value of Rice Monoculture Systems

Table 4 shows that the economic value of the rice–fish–duck farming ecosystem was the largest factor, followed by the ecological value, and the social value, which was the smallest. The proportional values of provisioning services, gas regulation, climate regulation, air purification, pest control, biodiversity maintenance, water regulation, soil conservation, soil organic matter accumulation, and cultural services were 53.10%, −0.58%, 34.00%, 3.62%, 0.15%, 0.37%, 0.09%, 0.02%, 2.28%, and 6.96%, respectively.

The biggest differences between rice monoculture and integrated rice–fish–duck farming were in the value of provisioning services, which increased by 208.66% due to the added value of fish and ducks, and their provision of animal protein to local people. Moreover, integrated rice–fish–duck farming increased the value of cultural services, especially in terms of tourism development, by 135.82%. Finally, the value of gas regulation increased by 54.41% due to integrated rice–fish–duck farming, which reduced greenhouse gas emissions by 14.70%. The integrated rice–fish–duck farming ecosystem could be considered an example of climate adaptation, which was consistent with the conclusions of previous studies. After the analysis, it was found that the net income of the integrated rice–fish–duck system was significantly higher than that of the rice monoculture system. The integrated farming system had higher comprehensive benefits, including increased income and indirect environmental protection value. Compared with the rice monoculture system, the construction of integrated farming systems in different ecological niches accelerates the absorption and metabolism of various substances in the terraces (Nie et al., 2020) [68]. Table 4 shows the values of each part of the system, which were obtained by performing conversions based on similar formulas.

## 4. Discussion

### 4.1. Comparison with the Ecosystem Service Value of Other Wetlands

One unit of terraced area equaled 247,971 CNY/hm^2^, while Zheng et al. (2018) [55] reported a value of 227,500 CNY/hm^2^ for a rice paddy in Yuanyang County. The value of the Honghe Hani Rice Terraces was slightly less than reported by Liu et al. (2020) [57], who reported a value of 255,529 CNY/hm^2^ for a rice-fish co-culture system in Ruyuan County, China. This was largely due to the differences in water regulation. Because Ruyuan County is a hilly area, the water storage and flood control capacity of the rice-fishing system was more obvious. The reported value of the plain area of the Sanyang wetland was 55,332 CNY/hm^2^ (Tong et al., 2007) [69]. The annual ecosystem value of the Ghodaghodi wetland in Nepal was estimated to be 0.67 million US dollars by Aryal et al. (2021) [70], but this study only considered wetland goods, tourism, irrigation, carbon sequestration, existence and bequest, and religious and cultural values. Zuze (2013) [71] found that the contribution of a wetland to the surrounding communities had an estimated annual value of 17.2 million USD in Malawi, which was also due to different value aspects, with the non-use value not estimated.

### 4.2. The Importance of the Ecosystem Service Values in the Hani Terraces

Among the different service values, the total production values of rice, fish, and especially duck products were the highest, accounting for nearly half (47%) of the total economic value, and could improve a farmer’s livelihood greatly. Sasmal et al. (2020) [72] found that Khaki Campbell ducks performed well in integrated rearing systems and laid 17.64% more eggs than the local duck breed (Pati duck). The provisioning service value therefore made the greatest contribution to the overall ecosystem service value. This was the largest difference between the present study and other published studies, such as Dai et al. (2022) [73] who found that the impacts of ecological factors on the ES supply were greater than the socioeconomic factors.

Climate regulation was the second largest (34%) contributor to the overall value, with an obvious cooling effect on terraced ecosystems, which has positive effects on the Hani terraces. This result was similar to previous studies of the Yuanyang Terraces, where climate regulation accounted for 35.2% of the overall ecosystem value (Zheng et al., 2018) [55]. However, Dai et al. (2021) [74], Liu et al. (2020) [57] and Liu et al. (2021) [64] found that climate regulation had a greater effect compared with the other service values. The absorption of heat and regulation of temperature play a role in alleviating the heat island effect, and the Hani-integrated farming system had a significant cooling effect in summer during periods of high temperature, consistent with the findings of Yoshida (2001) [75].

As a world cultural heritage site, the Hani terraces have beautiful scenery and large cultural service value, which accounted for the third part of the ecosystem value contribution. The Hani terraces have an important landscape aesthetic value (Wang and Marafa, 2021) [76]. The Hani people have interesting customs, production methods, and landscape design, and the layout of the terraces and villages is designed in sympathy with ecological and ethical values, creating a spectacular landscape and cultural tourism resource. The “Terraces on the Cloud·Dream of Red River” is becoming an increasingly popular brand. Through careful planning, based on the protection of this famous world cultural heritage, a sustainable tourist economy can be developed, highlighting the natural geographical conditions and local traditional culture, a fact also recognized by Tekken et al. (2017) [77] and Tilliger et al. (2015) [78].

The Hani terraces also provide an artificial wetland which improves the environment, retains water, reduces pesticide use, and maintains soil and biodiversity (McLaughlin and Cohe, 2013) [79]. The terraced field ecosystem acts to purify the air of dust, SO_2_, HF and NOx, which helps to improve air quality in the surrounding areas. The service value of environmental purification of the terrace ecosystem in the evaluation area was mainly a consequence of this air purification (3.62%). The Hani terraces contain a large volume of water, due to their unique water distribution pattern. The terraced wetlands at higher altitudes store water year-round, with a water depth of about 0.25–0.3 m. The terraces can maintain a certain volume of water throughout the year, and their soils have strong water retention and permeability properties, which play an important role in water regulation, as also demonstrated by Bai et al., 2016 [62]. Repair and reconstruction of the terraces and ditches has greatly increased their water storage capacity and conserved groundwater resources. Through predation, fish reduce the numbers and occurrence of pests and limit the damage they cause (Lansing et al., 2011) [80], so achieving an ecologically balanced farming practice is important. In the integrated rice–fish–duck farming system, microorganisms in the rice exchange nutrients with the soil (Zhang et al., 2018) [81] and the activities of aquatic animals and ducks help to loosen the soil, enabling more oxygen to penetrate and so promote the growth of rice. In addition, fish manure provides nutrients to fertilize the crop (Nayak et al., 2018) [82]. All of this can improve the soil structure. Fish and duck farming on the terraces has reduced the degree of soil compaction, increased soil bulk density and porosity, and improved soil nutrient and organic matter content. Integrated rice–fish–duck farming plays an important role in maintaining ecological balance, including support for more than 200 paddy weed species, a large number of invertebrates, and cranes and ibises (Zhang et al., 2016) [83]. Rice field ecosystems can enrich biodiversity, and further research on the biodiversity of these ecosystems is required.

Finally, we should point out that the value of O_2_ released by the terrace ecosystem in the evaluation area was greater than that of CO_2_ fixed, which is consistent with the study of Zheng et al., 2018, who focused on the Yuanyang terraces. However, the negative gas regulation value generated by the terrace field ecosystem needs further attention in order to make future changes to minimize its potential for harm.

### 4.3. Benefit Analysis of the Integrated Rice–Fish–Duck Ecosystem in the Hani Terraces

The integrated rice–fish–duck farming ecosystem in the Hani terraces not only protects the terraces but has also increased the income of farmers. Due to its unique cultural and geographical environment, as well as the ecological, replicable, and popular farming methods, the integrated rice–fish–duck farming ecosystem in the Hani terraces has a profound impact on the local economy, ecology, and society. The successful practice of integrated rice–fish–duck farming in the Hani terraces is a useful model for the sustainable development of agriculture and rural economies in other areas with an underdeveloped economy, fragile ecology, and rich culture (Liu et al., 2017; Bene et al., 2016) [84,85]. The purpose of this development was the same as in the Yangtze and Yellow River Basins (Fang et al., 2021) [86], where ecological protection and sustainable economic development were the main demands.

In terms of economic benefits, 13,373.33 hm^2^ of the rice–fish–duck ecosystem was effectively utilized as terraced field resources, improving the output rate of the Hani terraces, and increasing the economic income of farmers. After deducting the expenditures on seeds, labor, chemical fertilizers (manures), and feeding materials, an average profit of RMB 35,850/hm^2^ was achieved. The fundamental strategy for the protection of the Hani terraces is to increase farmers’ income. The economic returns for farmers continuing traditional agro-aquaculture production methods could be greatly increased and should encourage farmers to adopt the integrated rice–fish–duck farming system. The results were consistent with the study of Xu et al., 2021 [87], who found that conversion to rice–crayfish co-culture increased the net ES value by 145.3–176.9%.

In terms of the social benefits, the Hani terraces provide employment for fishermen, and their incomes have increased under the integrated rice–fish–duck ecosystem. The promotion and application of integrated rice–fish–duck continuous cropping in the Hani terraces has provided more employment for Hani women, who have received special training in farming technologies. Moreover, due to the development of tourism, there are more jobs for women and their family status has improved due to their increased income. This has contributed to family harmony, promoted better and faster development in remote areas, and assisted revitalization projects in rural areas. The integrated rice–fish–duck ecosystem will also ensure food security, promote the increase of grain production and farming incomes, protect the world cultural heritage site, and promote the development of tourism, these conclusions being consistent with those of previous studies (e.g., Sasmal et al., 2020) [71]. Honghe is rapidly becoming the center of tourism in Yunnan, driving the development of the local economy and increasing the income of local people (Zhang et al., 2018) [80].

In terms of ecological benefits, rice fields provide fish and ducks with an abundance of food sources and habitats. Fish and ducks feed on plankton, benthic organisms, aquatic insects, and microorganisms (Liu et al., 2021) [88] in the rice fields, thereby reducing the amount of feed required and saving energy. The swimming, feeding, and waste excretion of fish and ducks contributes to the weeding, fertilizing, and cultivation of rice, with the three processes being mutually beneficial and symbiotic. Organic fertilizers have been applied to inhibit the growth of weeds and loosen the soil structure, reducing the usage of chemical fertilizers and pesticides, and thereby reducing environmental pollution. This has resulted in the formation of a virtuous ecological cycle, with very significant ecological benefits.

Through the development of integrated rice–fish–duck farming, local people will obtain the opportunity to lead a meaningful life, with the efficient and fair use of resources and the environment. Measures need to be taken to effectively maintain the quality and quantity of environmental resources to achieve an equitable distribution, thereby contributing to sustainable well-being goals. The integrated rice–fish–duck farming ecosystem in terraced fields has encouraged farmers to change their production methods, inherit traditional agricultural systems, maintain the balance of the natural environment and social stability, and promote the sustainable development of an ecologically based socio-economic system (Liu et al., 2019) [88].

As more data becomes available, we are planning similar studies to measure and compare the ecosystem service values between high and low altitude environments.

## 5. Conclusions

In this study, a comprehensive ecological value assessment framework was applied to identify and evaluate the integrated rice–fish–duck farming ecosystem, in which the CICES classification standard played an important role. We determined the value of the integrated rice–fish–duck ecosystem of the Hani terraces from 10 different aspects. Some evaluations could not be obtained, and therefore, the ecosystem service values of the Hani terraces are likely to be higher in terms of actual production activities. The general service value of the integrated rice–fish–duck farming ecosystem in the study area was 3.316 billion CNY (equivalent to 510.09 million USD/hm^2^/year), of which the provisioning service value was 1.76 billion CNY (equivalent to 270.90 million USD/hm^2^/year), the regulation and maintenance service value was 1.32 billion CNY (equivalent to 203.76 million USD/hm^2^/year), and the cultural service value was 230.85 million CNY (equivalent to 3.55 million USD/hm^2^/year). The positive value of the ecosystem services of the Hani terraces in 2020 was about 513.07 million USD /hm^2^/year. Among all the positive assessed values, provisioning and temperature regulation were the largest, accounting for 53.11% and 34% of the total, respectively. The other values were ranked in descending order of culture (6.96%) > air purification (3.62%) > soil organic matter accumulation (2.28%) > maintaining biodiversity (0.37%) > pest control (0.15%) > water regulation (0.09%). The value of gas regulation was negative and accounted for −0.58% of the total ecosystem service value of the Hani terraces.

The sustainability of human society is based on the sustainability of ecosystems and their services. Through this investigation of the ecosystem values and market-oriented value of the rice–fish–duck system in the Hani terraces, it was calculated that the integration of traditional ecological models and modern agro-ecological techniques will have a huge impact. By implementing integrated rice–fish–duck farming, production, nutrition, income, environmental conditions, and livelihoods can be improved. The results of this study will further promote the protection of the regional ecological environment, the inheritance of cultural heritage, and the sustainable development of natural resources. This study will also improve public awareness of environmental protection and highlight the contribution of nature to the successful operation of agricultural systems, enabling the value of ecosystems to be correctly evaluated. The findings of the study will enable rapid and lasting progress to be made towards sustainable development for people and the environment. It is also an important reference for other countries and regions of the world with terraced fields, enabling them to achieve better production, nutrition, and livelihoods. Future research requires long-term observations and simulation experiments, and the field measurement of relevant data needs to further understand the internal mechanisms and evolution of the ecological services in terraced areas. This will promote the application of ecosystem service evaluations in policy and decision making.

## Figures and Tables

**Figure 1 ijerph-19-08549-f001:**
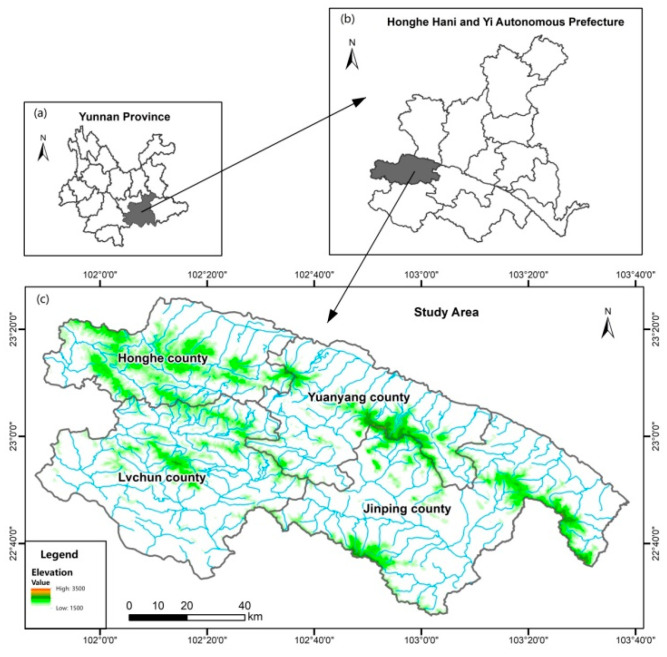
(**a**) Map of Yunnan province; (**b**) Map of Honghe Hani and Yi Autonomous Prefecture; (**c**) A map of study area showing topography with waterways.

**Figure 2 ijerph-19-08549-f002:**
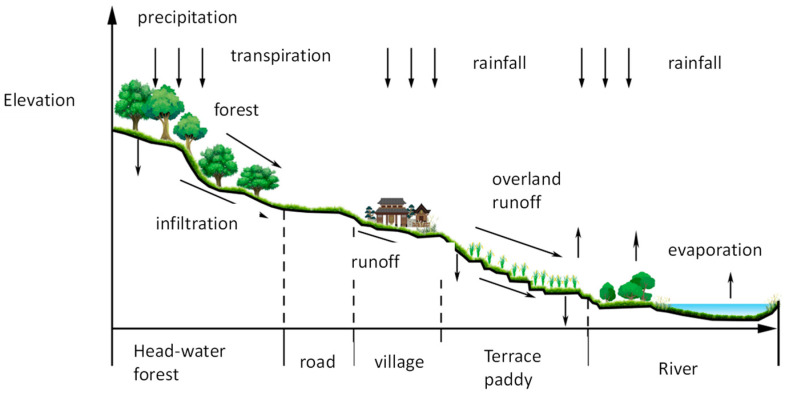
Forest-village-terrace field-water system.

**Figure 3 ijerph-19-08549-f003:**
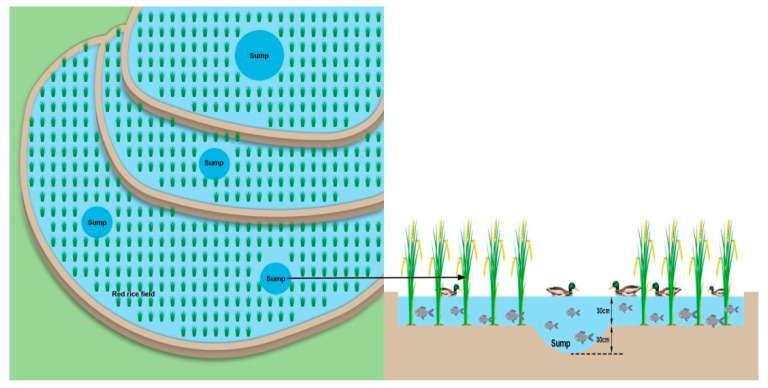
Integrated rice–fish–duck farming model in Honghe Hani terraced field.

**Figure 4 ijerph-19-08549-f004:**
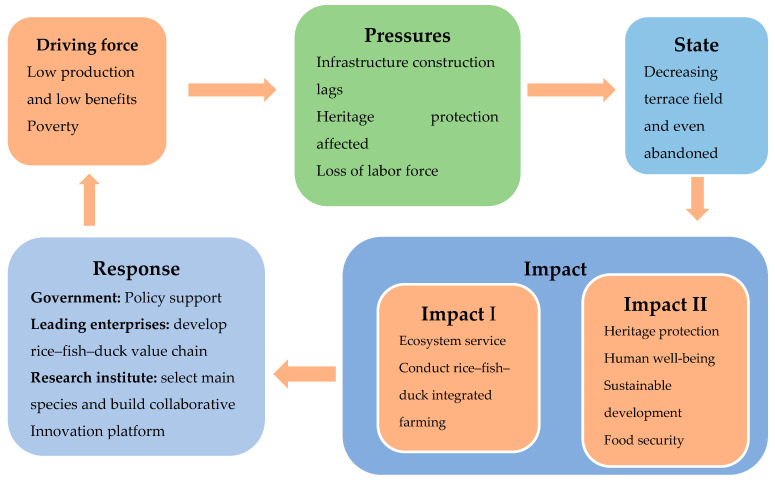
DPSIR model of the Hani rice–fish–duck integrated farming.

**Table 1 ijerph-19-08549-t001:** Required data.

Some monitoring data	evaluation area
	number of the growth periods of rice
	the number of days of standing water for rice
	the number of hot days in summer
	price of agricultural water and coal
Provisioning	Red rice yield and market price
	Common carp yield and market price
	Duck and duck egg yield and market price
Regulation and Maintenance	average emission fluxes of CH_4_, rice CO_2_, and soil CO_2_ from rice fields
	average daily water evaporation in terraced fields
	average SO_2_, NOx, HF, and dust concentrations absorbed by terraced fields
	the cost of SO_2_ removal, NOx removal, HF removal and dust removal
	pesticide costs/hm^2^
	soil water infiltration rate
	market price of agricultural water
	organic matter, total nitrogen, total phosphorus, and total potassium contents in the soil tillage layer
	cost of fertilizer
	soil thickness in the tillage layer
	Soil bulk density
	Biomass of straw and rice root
	Carbon content of Straw and rice root
	annual CO_2_ and CH_4_ emissions
	market price of organic matter calculated as pure carbon
Cultural	the number of visitors
	the total number of tourists throughout the year
	total tourism revenue

**Table 2 ijerph-19-08549-t002:** The values of the ecosystem services classification of the Hani terraces rice–fish–duck integrated faming system (according to CICES V5.1).

CICESV5.1 Section	Division	Group	Ecosystem Services of the Rice–Fish–Duck System	Goods and Benefits Valued Economically	Estimation Method
Provisioning	Biomass	Cultivated Plants, Reared aquatic and animals for nutrition, materials or energy	1. Red rice, fish and duck for nutrition	Provisioning service	market price method
Regulation & Maintenance	Regulation of physical, chemical, biological conditions	Atmospheric composition and conditions	2. Carbon dioxide fixation from photosynthesis	Carbon fixation and oxygen release	afforestation cost method and industrial oxygen
			3. Oxygen release from rice photosynthesis		
			4. Reduced greenhouse gas emissions	Greenhouse gas reduction	GWP-Global Warming Potentials
			5. Regulation of temperature and humidity, including ventilation and transpiration	Climate control	replacement costs method
			6. Rice absorbs SO_2_, HF, NOx, and dust	Air purification	replacement costs method
		Pest and disease control	7. Reducing pesticides and herbicides	Pest control	replacement costs method
		Lifecycle maintenance and habitat and gene pool protection	8. Increase of fauna diversity and micro-organisms	Biodiversity	equivalent factor method
		Water conditions	9. Recharging groundwater	Water storage and retention	replacement costs method
		Regulation of soil quality	10. Reducing land abandonment	Soil conservation	opportunity cost
			11. organic accumulation	Maintaining soil nutrients	replacement costs method
Cultural	Direct, in-situ, and outdoor interactions with living systems that depend on presence in the environmental setting	Spiritual, symbolic and other interactions with natural environment	12. Elements of living systems used for entertainment or representation	Development of tourism	simulated market approach
		Intellectual and representative interactions with natural environment	13. Cultural value and heritage		
			14. Characteristics of living systems that enable aesthetic experiences		

**Table 3 ijerph-19-08549-t003:** Values of the various ecosystem services of the Hani terraces rice–fish–duck integrated faming system.

Ecosystem Service Value	Equation	Explanation of Parameters
*V*_1_: The value of providing primary product functions in terraced rice–fish–duck ecosystems (CNY);	*V_1_* = *T**r* × *P**r* + *T**_f_* × *P**_f_* + *T**_d_* × *P**_d_*	*Tr*: The yield of rice in the evaluation area (t); *P**r*: Market price of rice in the evaluation area (CNY/t); *T_f_*: The yield of common carp in the evaluation area (t); *P_f_*: Market price of common carp in the evaluation area (CNY/t); *T_d_*: The yield of duck and duck egg in the evaluation area (individual); *P**_d_*: Market price of duck and duck egg in the evaluation area (CNY/ individual)
*V*2 represents the total value of greenhouse gas emissions	$V_{CO_{2}}\;$= *Tr* × *a* × *b* × *M_C_* × *C_afforestation_**_C_*	$V_{CO_{2}}$ represents the CO_2_ fixation service value; *a* is the economic coefficient of rice. In our study, *a* value of 0.5 was used (Liu et al., 2020) [57]. *b* is one rice field can fix 1.63 g of CO_2_ according to the photosynthesis equation; *Mc*: CO_2_ consists of 27.27% C; *C_afforestation_* was 327.32(CNY/t)
	$V_{O_{2}}\;$= *T**r* × *a* × *c* × *C_afforestation_**_O_*	$V_{O_{2}}$ represents the release O_2_ service value *c* is that produce 1.19 g of O_2_ for every gram of rice in the form of dry matter.
	$G_{CH_{4}}\;$= *A* × *D* × *g1*	$G_{CH_{4}}$ represents the total CH_4_ emissions (kg) *A* was the evaluation area of the integrated rice–fish–duck ecosystem (hm^2^); *D* was the growing period of rice (d). *g1* was the average emission flux of CH_4_ in rice field ecosystem in Yunnan Province(kg·hm^2^/d) (Zu, 2007) [58]
	$G_{CO_{2}}\;$= *A* × *D* × (*g2* + *g3*)	$G_{CO_{2}}$ represents the total CO_2_ emissions (kg) *g2* was the average emission flux of rice CO_2_ in the rice field ecosystem in Yunnan Province; *g3* was the average emission flux of soil CO_2_ in the rice field ecosystem in Yunnan Province (kg·hm^2^/d) (Zu, 2007) [58]
	$GT_{CO_{2}}\;$$=(G_{CO_{2}}\;$$+d\times \;G_{CH_{4}}$)/1000	$GT_{CO_{2}}$ represents the CH_4_ and CO_2_ emissions in the rice field ecosystem were converted into total CO_2_ emissions (t) *d* was the coefficient of conversion of CH_4_ to CO_2_
	$V_{2}=GT_{CO_{2}}\;$× *M_C_* × *C_afforestation_**_C_* (Bai et.al., 2017) [59]	
*V*3 represents value of climate regulation (CNY)	*Q_t_ = E* * × Tn*	*Qt* represents the total cooling effect (mm); *E* represents average daily water evaporation in terrace fields (mm/d), that is, 3.83 mm/day (Sun et al., 2006) [60] *Tn* represents the number of hot days in summer
	*V*3 = *Qt* × *A* × *e* × *P*	*e* represents the amount of heat consumed on evaporating 50 mm water in a 1 hm^2^ in terraces field is equal to the heat required to burn 30.57 tons of standard coal (Zhou et al., 2009) [61]; *P* represents the price of coal
*V*4 represents value of the air purification (CNY)	*V*4 = (*Q**d* × *P**d* + *Qs* × *Ps* + *Qh* × *Ph* + *Qn* × *Pn*) × *A*	*Qd*, *Qs*, *Qh Qn* and represents the average flux of dust, SO_2_, HF and NOx absorbed by terraced ecosystems (kg/hm^2^); *Pd*, *Ps*, *Ph*, *Pn* represents the cost of dust, SO_2_, HF and NOx removal (CNY/kg)
*V*5 represents value of the pest control (CNY)	*V*5 = *Q**p* × *A*	*Qp* represents the reduced pesticide costs per hm^2^ by integrated rice–fish–duck farming ecosystem
*V*6 represents value of the maintaining biodiversity (CNY)	*V*6 = *P_WC_* × *A*	*P_WC_* represents value of biodiversity in each unit of integrated rice–fish–duck farming ecosystem.
*V*7 represent the value of water regulation (CNY)	*V*7 = *f* × *A* × *P_W_* × *T**w*	*f* represents the soil water infiltration rate in terrace field (mm/d), that is 7.22 mm/d (Bai et al., 2016) [62]); *P_W_* represents market price of agricultural water, and that is 0.2 CNY/m^3^ according to the survey data. *Tw* represents the number of days of standing water during the growth period of rice (d)
*V*8 represents the value of soil conservation of Hani terraces (CNY)	*V*8 = *A* × *ST* × *SBD* × ∑(*SOM* + *TN* + *TP* + *TK*) *× P_F_*	*ST* represents soil thickness in the tillage layer; *SBD* represents soil bulk density; ∑(*SOM* + *TN* + *TP* + *TK*) represents sum of organic matter, total nitrogen, total phosphorus and total potassium content in soil; *P_F_* represents fertilizer price.
*V*9 represents value of soil organic accumulation;	*Isoc* = *Nt* × 5 × *Cr* + *Ns* × 11% × *Cs*	*Isoc* represents the soil organic matter input (kgC/hm^2^); *Nt* and *Ns* are the biomass of rice roots and straw, respectively (kg/hm^2^); *Cr* and *Cs* are the carbon content of rice roots and straw, respectively.
	*Osoc* = $R_{CO_{2}}\;$$\times 0.27\;+\;R_{CH_{4}}\;$*×* 0.75	*Osoc* represents the soil organic matter output (kgC/hm^2^); $R_{CO_{2}}$ and $R_{CH_{4}}$ are the annual CO_2_ and CH_4_ emissions from terrace fields, respectively; 0.27 and 0.75 are the coefficients for converting CO_2_ and CH_4_ to pure carbon.
	*Bsoc* = *Isoc* − *Osoc*	
	*V*9 = *Bsoc* × *Psoc*	*Psoc* is the market price of organic matter calculated as pure carbon (1.53 CNY/kg C) (Jiang (2016) [56])
*V*10 represents value of cultural service	*V*10 = *(Nrff/Nt)* × *Rz*	*Nrff* is the number of visitors to the rice-fish Culture Festival; *Nt* was the total number of tourists throughout the year according to the yearbook of Honghe; *Rz* was the total tourism revenue (CNY/year)

**Table 4 ijerph-19-08549-t004:** Comparison of the valuations of the integrated rice–fish–duck farming ecosystem and the rice monoculture system (2020).

Section	Ecosystem Services of the Rice–Fish–Duck System	Value (10 Thousand)	(%)	Rice-Monoculture System	Added Value	Added Proportion
Provisioning Service value	red rice, fish and duck	176,086.68	53.10%	43,088.88	132,997.8	208.66%
Gas regulation	carbon fixation and oxygen release	5230.84		5230.84	-	-
	greenhouse gas reduction	7166.86		8220.23	1053.37	14.70%
	gas regulation	−1936.02	−0.58%	−2989.391585	−1053.37	−54.41%
Climate regulation	cooling effect	112,737.17	34.00%	112,737.17	-	-
Air purification	purification of air quality	11,994.02	3.62%	11,994.02	-	-
Pest control	reduce the pesticide	501.50	0.15%	-	-	-
Maintaining biodiversity	maintaining ecological balance and biodiversity	1219.62	0.37%	1219.62	-	-
Water regulation	conserve of groundwater sources	308.98	0.09%	308.98	-	-
Soil conservation	improve the soil structure	58.5764	0.02%	60.36	1.78	2.96%
Soil organic accumulation	accumulation of organic matter	7562.99	2.28%	7562.99	-	-
Cultural service	landscape aesthetic value and tourism	23,085.33	6.96%	9789.28	13,296.05	135.82%
	CNY	331,619.5452	100%	183,771.9128	147,847.6324	80.45%

## Data Availability

In the results section, data supporting reported results can be found.

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
