# Peer review of "Valuation of Ecosystem Services for the Sustainable Development of Hani Terraces: A Rice–Fish–Duck Integrated Farming Model"

_ijerph, 2022, doi:10.3390/ijerph19148549_

Round 1
Reviewer 1 Report
I found your work interesting, the manuscript is well written, the results are clear and well presented. Simply put, I liked it. Only a minor spell check required of english language is required.
Author Response
Thank you very much for appreciating my article and thank you very much for your comments. I have checked and modified the spell of english language. Thank you very much again.
Reviewer 2 Report
Please see the attachment.

Reviewer 3 Report
The valuation of ecosystem services is an important issue in the Honghe Hani Rice Terraces. This study is well-done and developed a valuation model to assess the value of the ecosystem services of Hani terraces. I suggest some modifications before publication.
1. A few data of this research are obtained through questionnaire survey. It is suggested to attach the specific contents of the questionnaire survey in the appendix.
2. Ecosystem service and function are two different concepts. In the introduction and method part, function and service are used together for many times, such as line56, 57, 61. etc. It is suggested to distinguish the two concepts. The author can read the two papers to understand these two concepts: “Spatial assessment of ecosystem functions and services for air purification of forests in South Korea” “To Value Functions or Services? An Analysis of Ecosystem Valuation Approaches”
3. Figure 1 lacks legend. Is the green part rice?
4. There are too many contents in Table1,2,3, and the three segment line format makes the readers confused. It is suggested to have more dividing lines inside the table.
5. Line 159-161 referred that “A driving force, pressure, state, impact, and response (DPSIR) analysis model was established to evaluate the ecosystem service value of the integrated farming ecosystem”,But in line 278-280 referred that “we used the DPSIR model (Svarstad et al., 2008) to identify the driving factors and responses of the Hani terraces ecosystem as a rice-fish-duck system.” In this study, we did not find that the author did the analysis of driving factors and response. The value model is used to evaluate the value of ES, so how is DPSIR model combined with the evaluation value model in this paper? Some recent publications should be discussed, for example, Predicting the supply–demand of ecosystem services in the Yangtze River Middle Reaches Urban Agglomeration; Identifying the impacts of natural and human factors on ecosystem service in the Yangtze and Yellow River Basins; Assessing the ecological balance between supply and demand of blue-green infrastructure
6. 2.2 data collection: It is suggested that the data to be used should be listed in a table to more clearly indicate which data are used in this study.
7. Is the formula of Table 2 self-established or quoted by others? If it is quoted by others, please indicate the source. If you established it, please explain the basis.
8. How to determine the economic value, economic value and social value of line 572-574? Which service values belong to economic value, economic value and social value? What is the basis of this classification?
Round 2
Reviewer 2 Report
Please see the attachment

Author Response
Thanks you very much for your suggestions, I totally accept them. Your careful review has made our article better and professional.
- Thank you for carefully review. We have modifed the table name below 2.3.
- Thank you for carefully review. We have corrected the subscript spelling throughout the whole manuscript.
Reviewer 3 Report
The authors have revised this manuscript according to the comments from the reviewers
Author Response
Thank you very much for your very positive and encouraging comments.